# New Triple Metallic Carbonated Hydroxyapatite for Stone Surface Preservation

Lorena Iancu [1], Ramona Marina Grigorescu [1,*], Rodica-Mariana Ion [1,2,*], Madalina Elena David [1], Luminita Predoana [3], Anca Irina Gheboianu [4] and Elvira Alexandrescu [1]

[1] National Institute for Research & Development in Chemistry and Petrochemistry—ICECHIM, 202, Spl. Independentei, 060021 Bucharest, Romania; lorena.iancu@icechim.ro (L.I.); madalina.david@icechim.ro (M.E.D.); elvira.alexandrescu@icechim.ro (E.A.)

[2] Doctoral School of Materials Engineering Department, Valahia University of Targoviste, 35 Lt. Stancu Ion, 130105 Targoviste, Romania

[3] "Ilie Murgulescu" Institute of Physical Chemistry, Romanian Academy, 202, Spl. Independentei, 060021 Bucharest, Romania; lpredoana@yahoo.com

[4] Institute of Multidisciplinary Research for Science and Technology, Valahia University of Targoviste, 130004 Targoviste, Romania; anca@icstm.ro

* Correspondence: ramona.grigorescu@icechim.ro (R.M.G.); rodica_ion2000@yahoo.co.uk (R.-M.I.)

**Abstract:** This paper presents the synthesis of the triple substituted carbonated hydroxyapatite with magnesium, strontium and zinc (Mg-Sr-Zn-CHAp), as well as its structural, morphological and compositional characterization. The analytical techniques used (WDXRF, XRD and FTIR) highlighted, on the one hand, the B form for the apatite structure, as well as the presence of the three metal ions in the apatite structure, on the other hand (small shifts of 1120–900 $cm^{-1}$ and 500–600 $cm^{-1}$ absorption peaks due to the metals incorporated into the CHAp structure). The ratio between the metallic ions that substitute calcium and $Ca^{2+}$, and phosphorus is increased, the value being 2.11 in comparison with CHAp and pure hydroxyapatite. Also, by using imaging techniques such as optical microscopy and SEM, spherical nanometric particles (between 150 and 250 nm) with a large surface area and large pores (6 $m^2/g$ surface area, pores with 6.903 nm diameters and 0.01035 $cm^3/g$ medium volume, determined by nitrogen adsorption/desorption analysis) and a pronounced tendency of agglomeration was highlighted. Also, the triple substituted carbonated hydroxyapatite was tested as an inorganic consolidant by using stone specimens prepared in the laboratory. The efficiency of Mg-Sr-Zn-CHAp in the consolidation processes was demonstrated by specific tests in the field: water absorption, peeling, freeze–thaw behavior, chromatic parameters as well as mechanical strength. All these tests presented conclusive values for the use of this consolidant in the consolidation procedures of stone surfaces (lower water absorption, increased mechanical strength, higher consolidation percent, decreased degradation rate by freeze–thaw, no significant color changes).

**Keywords:** carbonated hydroxyapatite triple substituted; nanoemulsion method; stone specimen samples; consolidation of stone specimens





## 1. Introduction

The conservation of decayed stone-built heritage elements knew a continuous development during the last decades, mainly by developing consolidates with improved efficacy, substrate compatibility and durability in time [1–4]. A good consolidate is defined as a material applied on a degraded/fragile surface which leads to an increased sample's strength, without modifying its color and aspect. The performance of a consolidation treatment depends on several factors: the consolidate type, the applying procedures and concentration of consolidates, the substrate characteristics and the ambient conditions before, during and after the application [5–10].

Calcium phosphates, mainly hydroxyapatite—HAp ($Ca_{10}(PO_4)_6(OH)_2$, the most stable form at a pH of more than 4), were previously used as a stone consolidate. Although its

precipitation is influenced by the medium pH and the substrate nature, HAp improved the stone's strength and reduced the surface micro-cracks. HAp proved a better efficiency as a consolidation mineral than calcium oxalate (with solubility comparable with calcite and lattice incongruity with this substrate) [9,11–13].

Carbonated hydroxyapatite ($Ca_{10}(PO_4)_{6x}(CO_3)_x(OH)_2$—CHAp) can successfully substitute pure hydroxyapatite in many applications, including carbonated stone consolidation, especially for indoor objects [14]. It is a versatile material with high chemical stability, crystallinity, strength and solubility much lower than calcite (~0.0094 g/L compared to calcite ~0.014 g/L and HAp ~0.0003 g/L at 25 °C) [15,16]. CHAp can be classified into three types, depending on carbonate substitution: A-type CHAp when $CO_3^{2-}$ changes $OH^-$, B-type carbonate substitution when $CO_3^{2-}$ ions replace $PO_4^{3-}$ and AB-type, respectively when both ion types are replaced, being the most stable form of CHAp [17,18]. The apatite structure is modified due to the planar structure of carbonate ions and the weaker bond between $Ca^{2+}$ and $CO_3^{2-}$, compared with the bond between $Ca^{2+}$ with $PO_4^{3}$, thus decreasing the axial length a and increasing the axial length c [19].

In our previous research [10], carbonated hydroxyapatite and its metallic derivatives (Ag/Sr/Ba/K/Zn) were synthesized by the nanoemulsion method, and the resulting AB-type CHAp was used to obtain homogeneously and strengthen stone protective coatings. The inorganic materials showed good adhesion and homogeneity, leading to the higher mechanical strength of artificial stone specimens. Also, some selected consolidants, Ag-CHAp and Sr-CHAp, proved good consolidation efficacy when applied to the damaged area of some real samples with heritage value [20]. Under this context, double-substituted carbonated hydroxyapatite has been prepared and characterized [21]. A similar conformation of the apatite structure resulted from the simultaneous replacement of $Ca^{2+}$ with strontium and zinc, the obtained Sr-Zn-CHAp structure presenting an appropriate behavior in stone consolidation processes. Sr-Zn-CHAp improved the stone's mechanical strength and prolonged freeze–thaw resistance, without affecting the chromatic parameters of the surface. The flexible three-dimensional structure of apatite allows the insertion of a mixture of ions such as divalent ions ($Sr^{2+}$, $Zn^{2+}$, $Mg^{2+}$), monovalent ions ($Na^+$, $K^+$ and $Ag^+$) or even trivalent ions ($Bi^{3+}$, $Ln^{3+}$) in the calcium site. This ionic replacement influences the crystallite size, crystallinity, lattice parameters, morphology, particle size and shape, specific surface area, porosity and chemical, thermal and mechanical properties [17,22–24].

Before selecting this triple substituted carbonated hydroxyapatite, the similarity of this material (especially the pore values from literature) with its applications in the field of tissue engineering and bone grafts was taken into account. In comparison with the bone pore size that ranges between 300–500 μm [25], the lime mortar has pores between $10^{-8}$ m and $10^{-4}$ m ($10^{-2}$ to 100 mm) and the gypsum-rich mortars has capillary pores of $10^{-2}$–10 mm [26,27]. As the solubility of this new material is higher than HAp and CHAp, this material is proposed for interior wall surfaces, with controlled humidity and without extreme climatic conditions (wind, light, rain and snow). Under such context, the pore size, and the filling possibility of this triple substituted hydroxyapatite on different stone models, especially, have been considered.

The paper aims to study the synthesis of carbonated hydroxyapatite triple substituted with magnesium, strontium and zinc (Mg-Sr-Zn-CHAp) via the nanoemulsion method, the investigation of calcium substitution with the three considered metallic ions and testing the efficacy of Mg-Sr-Zn-CHAp as inorganic consolidant. For these consolidation tests, a gypsum-based mortar was used to obtain artificial stones, and the treatments' efficacy was investigated by water absorption test, hammer impact energy test, peeling test and the resistance to freeze–thaw cycles. Also, the chromatic changes obtained after treatment were analyzed. Consolidation of stone materials on site is very difficult to assess, so testing the efficiency of consolidants on substrates that can simulate their composition and environmental behavior after treatment is preferable in order to find the most compatible and durable solution, with the best results in stone consolidation.

## 2. Materials and Methods

The elemental analysis was carried out using a Rigaku ZSX Primus II wavelength dispersive X-ray fluorescence (WDXRF) spectrometer (Rigaku, The Woodlands, TX, USA), equipped with an X-ray tube with Rh anode, 4.0 kW power, with front Be window (30 μm thickness). The WDXRF results were analyzed using an EZ-scan combined with Rigaku SQX fundamental parameters software (Rigaku, Tokyo, Japan) which is capable of automatically correcting all the matrix effects, including line overlaps.

The X-ray diffraction of Mg-Sr-Zn-CHAp powder was employed with an X-ray diffractometer (Rigaku Corporation, Tokyo, Japan), in the range $2\theta = 10–90°$, with $1°/min$ scan speed, $0.01°$ step width, using Cu-K$\alpha$ radiation—1.5406 Å wavelength. The Mg-Sr-Zn-CHAp crystal structure, the phase's composition and the crystallites' sizes resulted from the recorded patterns refined with the Rietveld technique.

The interactions between components and the formation of Mg-Sr-Zn-CHAp were analyzed by Fourier transform infrared (FTIR) spectroscopy, the spectra being recorded in the 400–4000 $cm^{-1}$ range, using a Nicolet 6700 FT-IR spectrometer (Thermo Fisher Scientific, Waltham, MA, USA), with KBr pellet technique, at 4 $cm^{-1}$ resolution.

Mg-Sr-Zn-CHAp morphology was investigated using a FEI Quanta Inspect FEG Scanning Electron Microscope (FEI, Hillsboro, OR, USA) with 30 kV of accelerating voltage. The images were processed using the ImageJ 1.50 software and the particle diameter was measured.

The nitrogen adsorption/desorption analysis was performed for porosity determination at a temperature of 77 K, in the relative pressure range $p/po = 0.005–1.0$, using a NOVA2200e Gas Sorption Analyzer (Quantachrome, Boynton Beach, FL, USA). The data were processed using the Nova Win version 11.03 software. The specific surface area was determined by the Brunauer–Emmett–Teller (BET) method and the pore diameter and volume were evaluated from the desorption branch of isotherm based on Barrett–Joyner–Halenda (BJH) model.

**Experimental part**

Synthesis of carbonated hydroxyapatite triple substituted with Mg, Sr and Zn

The triple substituted carbonate hydroxyapatite with magnesium, strontium and zinc was synthesized by the nanoemulsion method, at room temperature, according to [10,21]. In the first step, $(NH_4)_2HPO_4$ (Scharlau, Spain) and $NH_4HCO_3$ (Scharlau, Spain) in aqueous solution under magnetic stirring. After adjusting the pH to 11 using 1 M NaOH (ChimReactiv SRL, Romania), a solution of Ca $(NO_3)_2 \cdot 4H_2O$ (Scharlau, Spain) in acetone was added, obtaining spontaneous emulsification, and by Ouzo effect in acetone–water, some spherical nanometric CHAp particles have been obtained [28]. Afterward, Mg $((NO_3)_2 \cdot 6H_2O$ ChimReactiv SRL, Romania, strontium $(Sr(NO_3)_2$, Honeywell Fluka, Germany) and zinc $(Zn(NO_3)_2 \cdot 6H_2O$, Lachner, Czech Republic) have been added at room temperature and mixed 2 h. The reaction product supported filtration and washing; after that, it was lyophilized overnight and then calcined at 900 °C for 4 h.

Preparation, treatment and testing of stone specimen samples

The effect of the three metallic ions on CHAp efficiency as stone consolidant was tested on stone specimen samples prepared from a mixture of gypsum with 5% lime, sand and water in a 2:1:2 ratio. The blend was mixed for 10 min at 300 rpm using a laboratory vertical mixer until proper homogenization. Then, cube-shaped artificial stone specimens were obtained by pouring the mixture into silicon molds of $40 \times 40 \times 40$ mm size. The samples were dried at 60 °C for 12 h in an oven and then, at room temperature, for 2 weeks. Further tests were made after the complete drying.

Three aqueous solutions of Mg-Sr-Zn-CHAp with concentrations of 0.1 g/L, 0.25 g/L and 0.5 g/L were prepared by sonication for 60 min at 40 °C. Then, the stone samples were treated with each solution by brushing (B) and spraying (S), 3 times on each face of the sample, to highlight the influence of the application procedure on the consolidation results. The treated dry samples were characterized by specific tests to demonstrate the strengthening action of Mg-Sr-Zn-CHAp.

The water absorption of treated samples was determined, according to STAS 6200/12-73 [28], based on Equation (1), after the dried sample's immersion for 24 h in distilled water:

$$\text{Water absorption (\%)} = [(W_2 - W_1)/W_1] \times 100, \tag{1}$$

where $W_1$ is the mass of the sample dried at 40 °C for 8 h and $W_2$ is the mass of the same sample immersed in water for 24 h.

The compressive strength was estimated by hammer impact energy test, using a Silver Schmidt Proceq apparatus, type L, with 0.735 Nm impact energy and strength range of 10–100 N/mm$^2$, according to ASTM C805 [29]. Ten measurements were recorded for each stone sample. The estimated compressive strength, expressed in MPa, was calculated based on the recorded rebound number (Q), using the apparatus constant (2.77), in Equation (2). The reference equation of the apparatus was tested previously for this mortar composition, with correlated results. No significant influence of a potential hard surface on the estimated compressive strength as a bulk property was observed.

$$\text{Compressive strength} = 2.77 \times e^{0.048 \times Q}. \tag{2}$$

The peeling test was performed in order to assess the cohesion of the synthesized consolidant on the surface of artificial stone samples, according to Drdácký et al. method [30], by using Scotch Cristal tape (3M) with 10 repetitions over the same position. After approximately 90 s of application with constant pressure (2 kgf/cm$^2$), the tape of $1 \times 1.25$ cm dimensions was removed steadily, at a 10 mm/s approximative rate and at 90° angle. After weighing the tape, the percentage of consolidation (% C) was calculated according to Equation (3) [31], considering the amount of detached material on untreated samples (TRM$_{untreated}$, g) and the amount of the removed material on treated artificial stones (TRM$_{treated}$, g):

$$\% \text{ C} = (\text{TRM}_{untreated} - \text{TRM}_{treated})/\text{TRM}_{untreated} \times 100. \tag{3}$$

The resistance of treated artificial stone samples to frost weathering was analyzed by the artificial freeze–thaw test, according to SR EN 12371:2010 [32], with adapted conditions to the samples' dimensions. The samples were dried at 105 ± 5 °C for 1 h (weighted, $m_1$, g), immersed for 15 min in distilled water at room temperature (saturated samples were weighted, $m_2$, g), then introduced into the freezer for 3 h at a temperature of −18 ± 5 °C and finally thawed for 1 h in distilled water. Each sample was subjected to 10 cycles of freeze–thaw, weighted after the last cycle ($m_3$, g), and the mass losses were expressed as the gelivity coefficient was calculated using Equation (4).

$$\% \ \mu_g = (m_2 - m_3/m_1) \times 100. \tag{4}$$

The colorimetric test was carried out using a CR-410 colorimeter (Konica Minolta, Japan). The differences between the $L_x^*$, $a_x^*$ and $b_x^*$ chromatic parameters of simulated stone samples after applying the Mg-Sr-Zn-CHAp suspensions and the non-treated sample were analyzed. The total color difference, $\Delta E_x$, was calculated using Equation (5), according to standard D 2204 [33]:

$$\Delta E^*_x = (\Delta L_x^{*2} + \Delta a_x^{*2} + \Delta b_x^{*2})^{1/2}, \tag{5}$$

where $\Delta L_x^*$—the difference in lightness, $\Delta a_x^*$—the chromatic deviation of a coordinate (red and green color) and $\Delta b_x^*$—the chromatic deviation of b coordinate (yellow and blue color) calculated for the treated stones, in comparison to the control values.

## 3. Results and Discussion

X-ray fluorescence, being a non-destructive characterization technique, is usually used to analyze the chemical structure and elemental composition of materials. The WDXRF

results for Mg-Sr-Zn-CHAp are presented in Figure 1 and Table 1. The partial replacement of $Ca^{2+}$ in the CHAp structure with zinc, strontium and magnesium ions can be highlighted by their presence in the apatite structure, with the reduction of $Ca^{2+}$ concentration. In order to put into evidence more precisely the Ca/P ratio and the structure of the newly synthesized compound, two insets were inserted in Figure 2, obtained by magnifying the specific area of the WDXRF spectrum: "inset A" for Ca and P evidence (100–150 degrees) and "inset B" (20–50 degrees.) for Mg evidence.

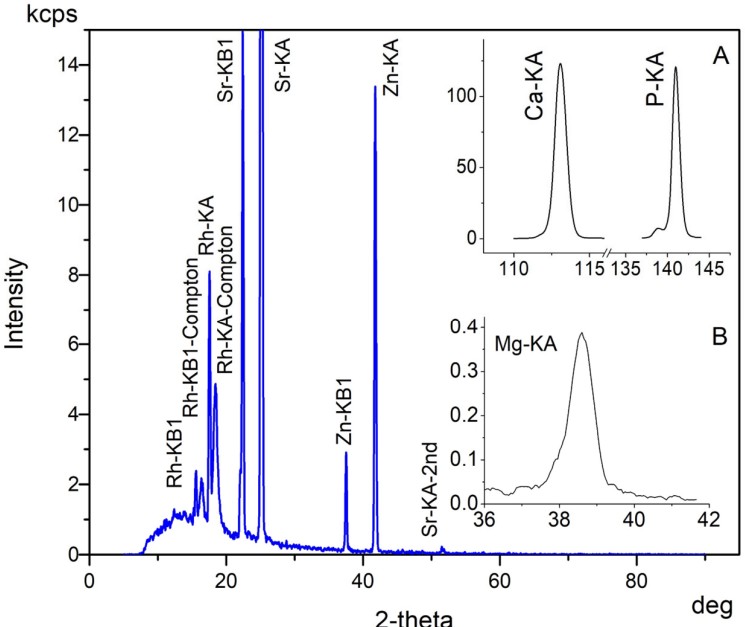

**Figure 1.** WDXRF qualitative analysis for identifying the main elements from Mg-Sr-Zn-CHAp (the Rh present lines are the characteristic elemental lines of anode photons). Inset A: Ca and P; Inset B: Mg.

**Table 1.** WDXRF elemental analysis of Mg-Sr-Zn-CHAp in comparison with CHAp.

| Element | Mass % | | Detection Limit | | Element | Intensity | |
|---|---|---|---|---|---|---|---|
| - | Mg-Sr-Zn-CHAp | CHAp | Mg-Sr-Zn-CHAp | CHAp | line | Mg-Sr-Zn-CHAp | CHAp |
| Mg | 0.3831 | - | 0.01665 | - | Mg-KA | 0.3626 | - |
| P | 18.1223 | 18.2630 | 0.02056 | 0.01957 | P-KA | 119.1271 | 130.2749 |
| Ca | 34.7967 | 35.8079 | 0.02689 | 0.02995 | Ca-KA | 122.7283 | 131.8156 |
| Zn | 1.2898 | - | 0.00829 | - | Zn-KA | 13.2132 | - |
| Sr | 1.7942 | - | 0.00473 | - | Sr-KA | 59.8188 | - |
| Si | - | 0.0136 | - | 0.00653 | Si-KA | - | 0.0399 |
| Na | - | 1.9411 | - | 0.04598 | Na-KA | - | 0.0399 |

According to the literature, $Ca^{2+}$ substitution into the apatite structure affects the Ca/P ratio which for CHAp is 1.96, higher than 1.67, characteristic of the stoichiometric hydroxyapatite structure (non-carbonated). According to [34], the double-substituted CHAp with $Ca^{2+}$ replaced by $Mg^{2+}$ and $Zn^{2+}$ is characterized by a Ca/P ratio of 1.95. For Mg-Sr-Zn-CHAp, the Ca/P value increases more, up to 2.11, and can be considered a confirmation of the substitution of calcium into CHAp structure with Mg, Sr and Zn [10,35].

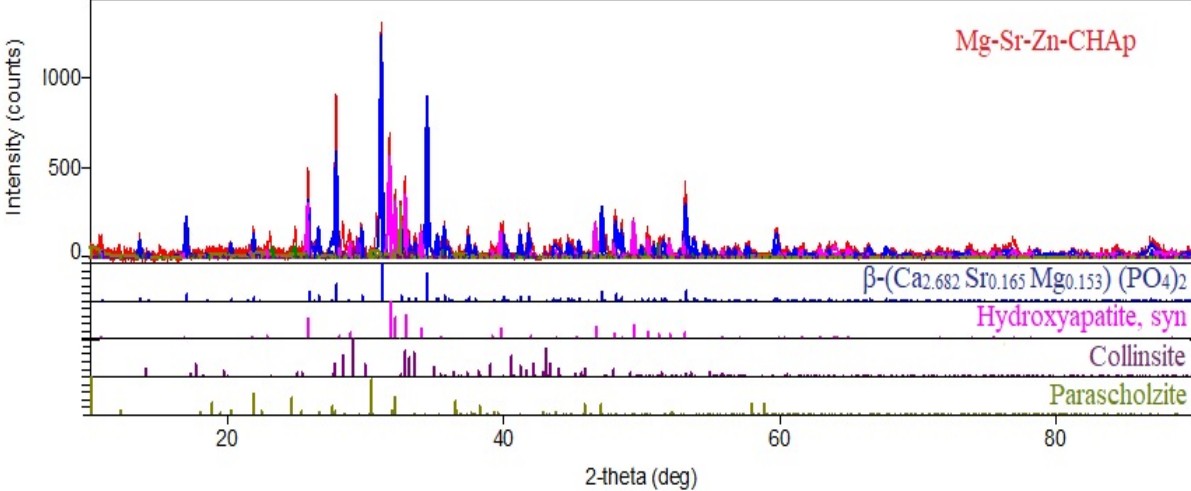

**Figure 2.** XRD diffractogram for Mg-Sr-Zn-CHAp, accompanied by the Rietveld refining diagram.

The XRD analysis of Mg-Sr-Zn-CHAp with $2\theta = 10$–$90°$ is presented in Figure 2. The specific peaks of CHAp are identified at $2\theta$ between 25 and $40°$ [10,35–37]. The lattice parameters, the crystallite size and the powder phases were determined. The incorporation of metal ions into the structure of carbonate hydroxyapatite was evidenced by identification of specific minerals using the Rietveld refinement, hydroxyapatite—DB card number 00-064-0738, substituted calcium phosphates ($\beta$-$(Ca_{2.682}\,Sr_{0.165}\,Mg_{0.153})\,(PO_4)_2$—DB card number 01-080-2757, collinsite $Ca_2Mg(PO_4)_2\,(H_2O)_2$—DB card number 04-013-3798 and parascholzite $CaZn_2(PO_4)_2\,2H_2O$—DB card number 00-035-0495) (according to ICDD-International Centre for Diffraction Data).

The crystallite size for Mg-Sr-Zn-CHAp hexagonal lattice (a = b and c) was calculated from XRD data, using Scherrer's Equation (6), and the lattice parameters were determined through the (*hkl*) peaks position of the apatite from XRD pattern according to Equation (7). The (002) peak was used for calculation.

$$L = (K \times \lambda)/(FWHM \times \cos\theta), \qquad (6)$$

$$1/d^2 = 4/3 \times (h^2 + hk + k^2)/a^2 + l^2/c^2, \qquad (7)$$

where L—crystallite size, in Å; K—the shape factor (0.9); $\lambda$—X-ray wavelength, 1.5406 Å; FWHM—the full-width at half-maximum of the peak, in radians; $\theta$—Bragg angle.

The Mg-Sr-Zn-CHAp specific peaks for each phase are mentioned in Table 2. By comparing the new triple substituted carbonated hydroxyapatite, with the single-form HAp, some small shifts can be observed [10]. The peak from $25.879°$ (0,0,2) for CHAp is evidenced at $25.787°$ and has a full width at half maximum two times smaller. The inferior values of FWHM indicate a higher crystallite size and a smaller degree of distortion brought by the metallic ions into the crystal structure [38].

The overall diffraction patterns were in agreement with the current literature [39–42] and showed the characteristic peaks of HAp derivatives. Table 2 lists and compares the significant peaks of the synthesized Mg-Sr-Zn-CHAp sample with the Joint Committee on Powder Diffraction Standards (JCPDS), no. 09-0432 for stoichiometric substituted CHAp, in good agreement with the literature [43]. The hydroxyapatite structure contains 10 calcium ions: 6 Ca (II) and 4 Ca (I), which are distributed among the two sites [22,44,45]. Different metal cations can substitute calcium and occupy the Ca (I) and/or Ca (II) positions. The resulting arrangement within the apatite lattice is strongly correlated to ionic radius value [38]: a cation with a larger radius than $Ca^{2+}$ (0.99 Å) tends to occupy site (II), site Ca (II) being bigger in volume than site Ca(I) [44]. $Sr^{2+}$ with a higher ionic radius, of 1.12 Å, occupies Ca (II) sites and the two smaller io ns, $Zn^{2+}$ (0.74 Å) and $Mg^{2+}$ (0.71 Å) substitute the Ca (I) positions, in good correlation with XRF results. In order to determine

the unit cell parameters of the substituted HA samples, the models previously reported by Dickens et al. [46] and Sudarsanan [47] were used as the starting conditions for the Rietveld refinement, which showed an excellent match to those input from the literature references.

**Table 2.** Results of the XRD diffraction for the main phases identified at Mg-Sr-Zn-CHAp.

| Identified Phase | Peaks/Miler Index | a, Å | b, Å | c, Å | Unit Cell, Å$^3$ |
|---|---|---|---|---|---|
| $\beta$-($Ca_{2.682}$ $Sr_{0.165}$ $Mg_{0.153}$) $(PO_4)_2$ | 17.025 (1,1,0) <br> 25.787 (1,0,10) <br> 27.808 (2,1,4) <br> 29.645 (3,0,0) <br> 31.101 (0,2,10) <br> 34.413 (2,2,0) | 10.424 | 10.424 | 37.19 | 3500 |
| Hydroxyapatite $Ca_{10}(PO_4)_6(OH)_2$ | 25.787 (0,0,2) <br> 31.101 (2,1,1) <br> 32.094 (1,1,2) <br> 32.828 (3,0,0) <br> 33.969 (2,0,2) <br> 39.771 (1,3,0) | 9.421 | 9.421 | 6.9 | 576 |
| Collinsite $Ca_2Mg(PO_4)_2(H_2O)_2$ | 17.025 (1,0,0) <br> 25.787 (1,1,0) <br> 28.310 (0,2,0) <br> 28.780 (1,0,1) <br> 41.783 (0,1,2) <br> 44.890 (2,0,1) | 5.757 | 6.716 | 5.431 | 186 |
| Parascholzite $CaZn_2(PO_4)_2 \cdot 2H_2O$ | 26.527 (0,2,1) <br> 27.808 (5,1,0) <br> 32.094 (1,1,2) <br> 33.534 (5,1,1) <br> 39.771 (2,2,2) <br> 47.368 (2,4,0) | 17.82 | 7.92 | 6.36 | 881 |

Hydroxyapatite is monoclinic, with a = 9.421 Å, b = 2a, c = 6.881 Å and $\gamma$ = 120° and at a temperature of 250° C, it passes into hexagonal symmetry, having network parameters, with a = b = 9.432 Å, c = 6.881 Å and $\gamma$ = 120° [48]. CHAp has a hexagonal structure, with a = b = 9.431 and c = 6.891 [10], while the Mg-Sr-Zn-CHAp sample has the following calculated parameters of the hexagonal crystal lattice: a = b = 9.421 Å and c = 6.9 Å. Changes in lattice parameters indicate a weaker bond between calcium and carbonate ions compared with the bond between calcium and phosphate, probably due to the inclusions of the metallic ions that lead to stabilizing the substituted $\beta$-TCP forms. The c/a ratio for HAp phase of Mg-Sr-Zn-CHAp is 0.73241 in comparison with non-substituted CHAp when the ratio is 0.73067, so the B-form of apatite is maintained [49]. In accordance with the FWHM decrease, the crystallite size of Mg-Sr-Zn-CHAp is 267 Å, with about 17% being higher than for CHAp (229.50 Å, according to our previous results [10]) due to the three metallic ions present in the structure. The combined effect of $Mg^{2+}$, $Sr^{2+}$ and $Zn^{2+}$ also influences the cell parameters of the more stable calcium phosphate phase. In a comparison of the a and c values of 10.427 Å and 37.451 Å for $\beta$-TCP, the Mg-Sr-Zn-CHAp substituted tricalcium phosphate has lattice parameters of 10.424 Å and 37.19 Å; thus, a unit cell of 3500 Å$^3$ resulted, compared to statistic value of 3527.2 Å$^3$ [50–53]. All these results, correlated with the literature studies on triple substituted apatite [35,51,54], can be considered proof of the successful preparation of this compound by the nanoemulsion technique.

The characteristic functional groups of the carbonated hydroxyapatite triple substituted with magnesium, strontium and zinc were evidenced by Fourier transform infrared spectroscopy (FTIR) in the 4000–400 cm$^{-1}$ range, as in Figure 3. As can be observed, the hydroxyapatite functional groups ($PO_4^{3-}$, $OH^-$ and $CO_3^{2-}$) are identified in the newly synthesized Mg-Sr-Zn-CHAp sample. The phosphate group is marked by the 472 cm$^{-1}$

absorption band, the intense bands from 1049 cm$^{-1}$ and at 976 cm$^{-1}$ ($\nu_3$ and $\nu_1$ vibration of P-O bond) and the two bands at 550–570 cm$^{-1}$ and 600–610 cm$^{-1}$ ($\nu_4$ bending vibrations of O-P-O groups). The OH-stretching and vibrational characteristic bands appear at 3200–3600 cm$^{-1}$ and at 1600–1700 cm$^{-1}$ [55].

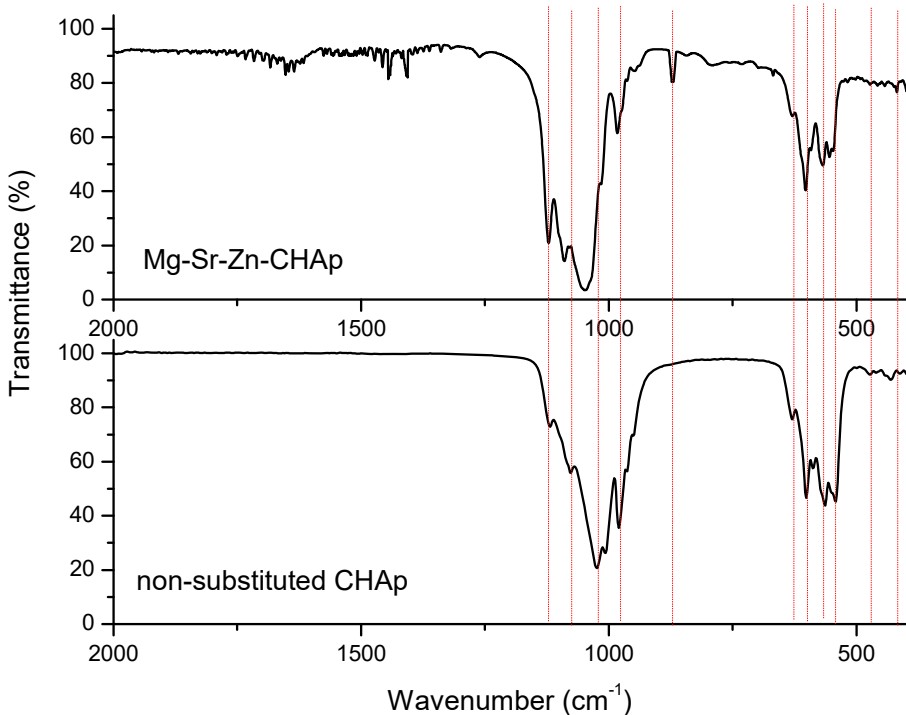

**Figure 3.** FTIR spectrum of Mg-Zn-Sr-CHAp and CHAp (the specific wavenumbers for both compounds are red-marked).

The FTIR spectrum confirms the B-form of apatite structure by the presence of the specific absorption bands of the carbonate group in the ranges: 870–875 cm$^{-1}$ for the $\nu_2$ bending mode, 1410–1430 and 1450–1470 cm$^{-1}$ for the $\nu_3$ stretching mode, according to literature [35,55]. $CO_3^{2-}$ bands are more intense as compared to non-substituted CHAp, suggesting a more stable apatite by a more efficient carbonation process. The ions Sr$^{2+}$, Mg$^{2+}$ and Zn$^{2+}$, which substituted Ca$^{2+}$ into the CHAp lattice, induce small differences in the form, position and intensity of peaks in the 1120–900 and 500–600 cm$^{-1}$ due to the incorporation of the metal ions [56,57].

The morphology of Mg-Sr-Zn-CHAp powder was evidenced by SEM micrographs at different magnifications, as presented in Figure 4. The sample, as can be observed, has a grain structure. At higher magnification, spherical particles of approximately 150–250 nm can be observed, which confirms the preliminary hypothesis regarding its use for the capillary pore network of some gypsum mortars. The tendency to agglomerate, as already observed for double-substituted CHAp with magnesium/strontium and zinc [21,34], lead to some cluster formations, probably due to the weak physical interactions between the particles.

The presence of Sr, Mg and Zn substitutions researched in this study did not influence the particle morphology [58,59]. Suchanek et al. depicted similar acicular large, agglomerated morphologies of Mg-HAp, previously. Ren et al. described typical particle shapes without sharp face angles for Zn HA [60]. In all cases, SEM micrographs showed that the crystallites of the investigated samples were aggregated resulting in the formation of nano-sized particles and additionally larger agglomerates as well. A uniform agglomeration of irregular nanostructures could be observed which is known to differ based on the co-doping ion chosen [61].

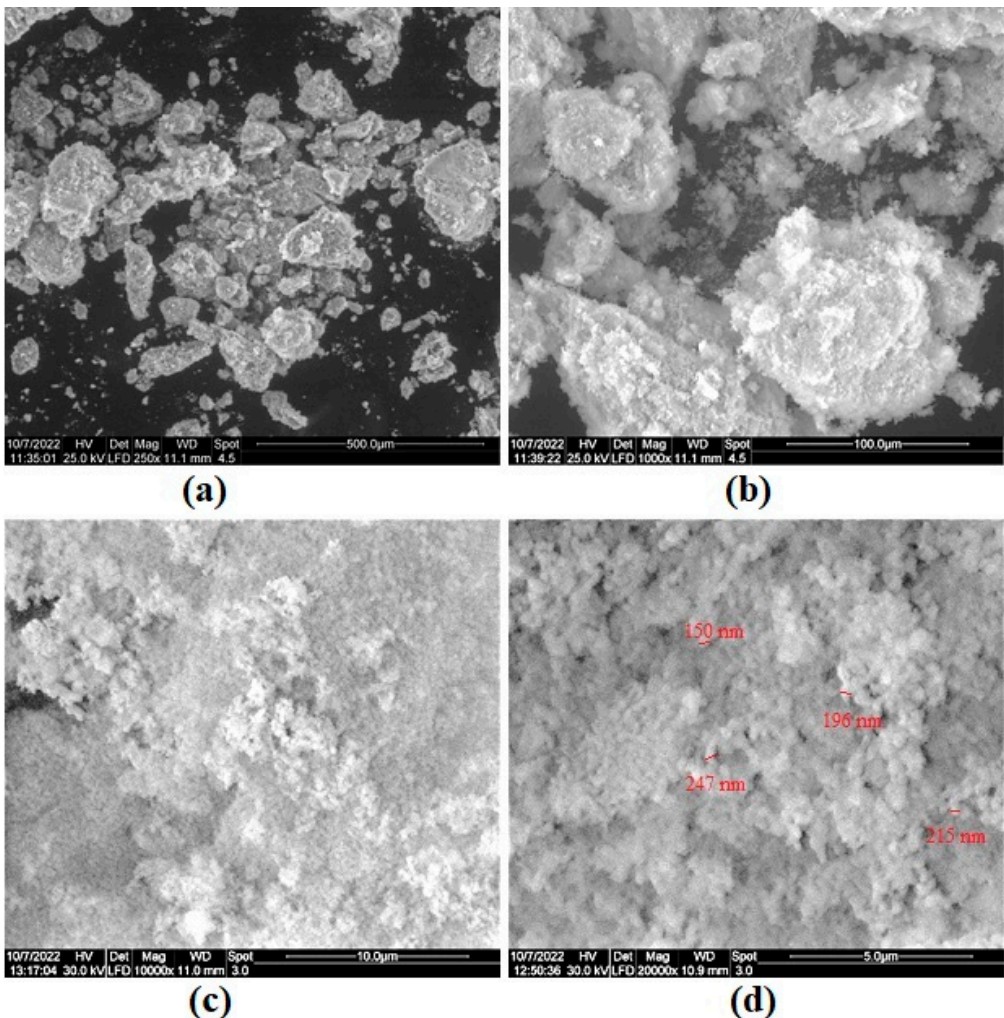

**Figure 4.** SEM micrographs of the Mg-Sr-Zn-CHAp powder at different magnifications. (**a**) 250×; (**b**) 1000×; (**c**) 10,000× and (**d**) 20,000×.

The surface of this triple substituted carbonated hydroxyapatite appeared to have a granular, nano-topography, likely to improve the overall porosity and surface area. The pore structure was analyzed by the nitrogen adsorption–desorption isotherms. Mg-Sr-Zn-CHAp presented a 6 $m^2$/g surface area and pores with 6.903 nm diameters and 0.01035 $cm^3$/g medium volume. The values are higher than our previous synthesized carbonated hydroxyapatite double substituted with strontium and zinc [21], which had a specific surface area of 2.723 $m^2$/g, pore diameter of 2.946 nm and pore volume of 0.05 $cm^3$/g. So, the pore structure of this material is modified by introducing a third metallic ion into the CHAp structure. Larger pores with higher surface area are thus obtained, in accordance with the smaller value of the crystallite size [55].

In conclusion, the presence of the three metallic ions in carbonated hydroxyapatite obtained by the nanoemulsion technique is confirmed and spherical particles, with high surface area and pores as it has been evidenced.

The synthesized Mg-Sr-Zn-CHAp powder has been used as a consolidant agent in aqueous suspension for stone specimen samples prepared in the lab. The consolidation capacity of Mg-Sr-Zn-CHAp on stone specimens was studied by water absorption test, hammer impact energy test, scotch tape test, freeze–thaw test and colorimetric analysis.

Mg-Sr-Zn-CHAp forms a consolidation coating when applied on the stone specimens' surface, as can be observed by microscopic analysis, Figure 5a,b, calculated with ImageJ software (free version).

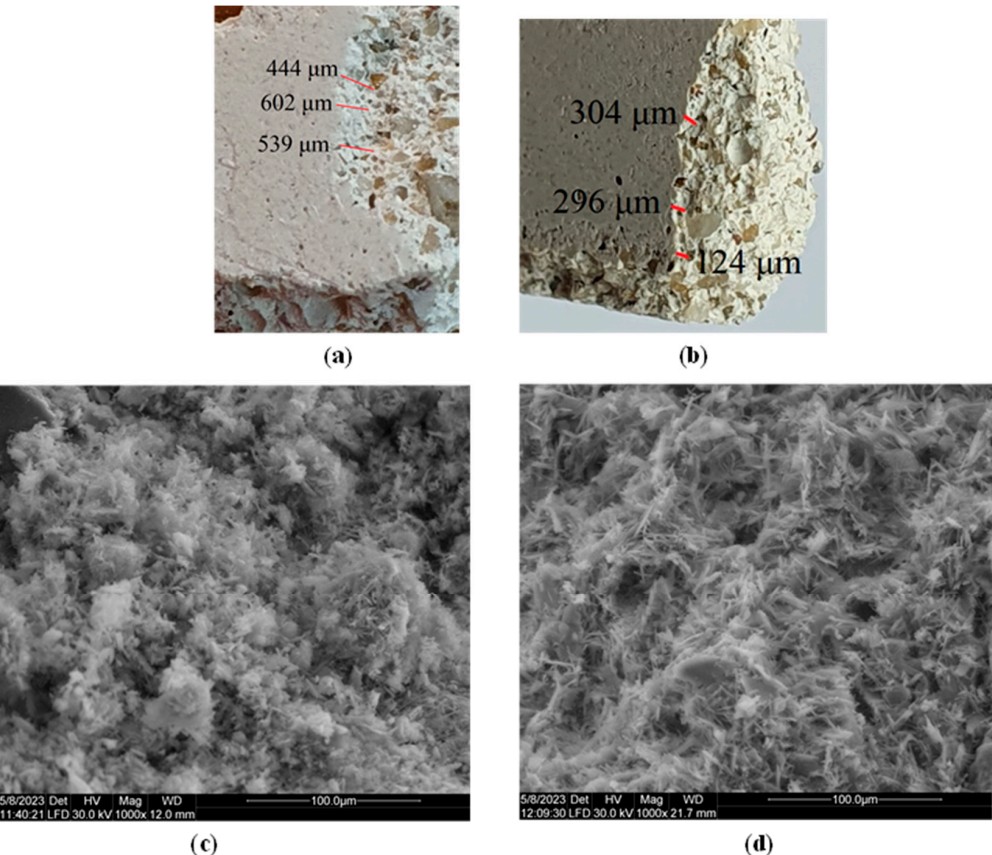

**Figure 5.** The consolidation layer was obtained by 0.25 g/L Mg-Sr-Zn-CHAp application by brushing (**a**) and spraying (**b**) and SEM micrographs of superficial (**c**) and depth (**d**) part of 0.25 g/L brushed specimens.

A more compact superficial coating layer is obtained when the small particles fill the stone pores. This layer of 400–600 μm for brushing and 120–300 μm for spraying, both at 0.25 g/L concentration, could be evidenced by optical microscopy as the abundance of Mg-Sr-Zn-CHAp is higher at the treated surface and decreases along the specimen depth. This phenomenon was also observed for other consolidates (e.g., hydroxyapatite generated from diammonium hydrogen phosphate [6,62], applying by brushing a minimum of 10 times), even if the several centimeters penetration was observed by other techniques. Thus, SEM micrographs were achieved from the specimen surface and at a depth of 1.5 cm of these specimens, as shown in Figure 5c,d. The more compact surface layer achieved by filling and covering the pores of the sample can further influence the stone's durability and environmental exposure by controlling the absorption of water into the capillary structure of the material and the related mechanical properties. In the depth of the treated sample, the acicular aspect of gypsum [27] is predominant, but the consolidation process is uniformly achieved, even in the pores volume.

Thus, the water absorption of Mg-Sr-Zn-CHAp treated samples, both by brushing and spraying, was determined in order to evaluate the stone deterioration in environmental conditions and to monitor the effect of the consolidation treatment. The values are represented in Figure 6a. Modifying the porosity of the tested stone specimens with particles of high surface area creates a barrier to water penetration into the substrate, by blocking the capillaries of the transport system. By increasing the consolidate concentration, the water absorption proportionally decreases, the smallest values being obtained for the 0.5 g/L concentration. The best results are obtained when Mg-Sr-Zn-CHAp is applied by brushing, taking into consideration also the dimensions of the superficial layer formed after consolidating the simulated stone samples.

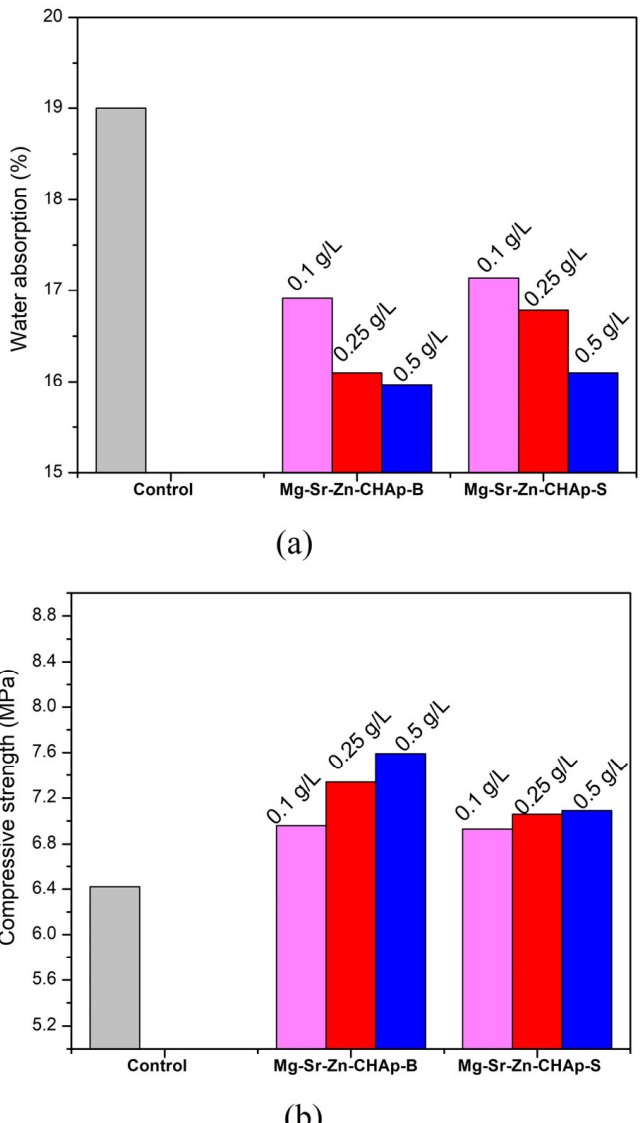

**Figure 6.** (**a**) Water absorption and the (**b**) mechanical compressive strength determined for control and samples treated with Mg-Sr-Zn-CHAp. (The Control sample is grey marked).

The mechanical compressive strength was estimated by measuring the rebound energy of the consolidate samples using Mg-Sr-Zn-CHAp, as shown in Figure 6b. The synthesized consolidant increases the mechanical strength value by 8%–18%, depending on the application method and the concentration of Mg-Sr-Zn-CHAp solution. Even the spraying is a faster method, it is less precise and does not allow monitoring of the applied consolidant amount [63], making it less efficient. Thus, in accordance with the published results [10,15,21,64,65] that proved efficient consolidation of stones by apatite treatment, especially CHAp-based materials, it can be stated that substituting calcium with the three metallic ions improves the consolidation process more.

The peeling test, the so-called scotch tape test, is used for testing the cohesion of consolidate to the stone substrate [66] After applying the pre-formed tape on the surface and pressing with constant pressure, the removed tape was weighted and the amount of detached material was calculated as the difference between the initial tape and after peeling, as shown in Table 3. The resulting value can give strong information regarding the binding strength between the two materials. Approximately, 7 mg were detached from the untreated simulated stone sample. The peeling test results indicate that the superficial

cohesion on the treated stone samples is improved as the amount of detached material is smaller by ~40%–50% compared to the control sample.

**Table 3.** Amount of the detached material on the scotch tape of 1.25 cm$^2$ and the calculated consolidation percent.

| Sample | | Material Detached, mg | Consolidation, % |
|---|---|---|---|
| Untreated | | 7.11 | - |
| CHAp Brushing | 0.25 g/L | 3.33 | 53.16 |
| Mg-Sr-Zn-CHAp Spraying | 0.1 g/L | 3.45 | 51.48 |
| | 0.25 g/L | 3.36 | 52.74 |
| | 0.5 g/L | 3.75 | 47.26 |
| Mg-Sr-Zn-CHAp Brushing | 0.1 g/L | 3.03 | 57.38 |
| | 0.25 g/L | 2.92 | 58.93 |
| | 0.5 g/L | 2.98 | 58.09 |

This Scotch Tape test has been applied in the case of nanolime in isopropanol applied by brush on Lecce stone followed by DAP application by brush. The nanolime + DAP application enhanced hydric properties, surface cohesion and strength of the treated stone to the extent that it can be suggested for in situ applications [67].

Also, the same test has been applied for Nanorestore, and a correlation with the salt crystalization has been adopted, with an increase in the crystallization pressure within the pores being possibly responsible for stone decay [68].

The consolidation percent is higher compared to CHAp and it is influenced by the concentration of applied solutions. The values are directly proportional with 0.1 g/L and 0.25 g/L concentration, but decreases at 0.5 g/L, probably because Mg-Sr-Zn-CHAp has the tendency to form agglomerates with higher dimension than the stone pores. As expected, increased cohesion and thus a more efficient consolidation is assessed when Mg-Sr-Zn-CHAp is applied by brushing, as compared to sprayed samples.

Frost resistance is another important factor for the durability of the building stone, knowing that the freeze–thaw cycles of the water inside the stone pores induce some internal stresses, leading to cracking and progressive desegregation of material [69]. However, it should be noticed that many stones have deteriorated for a greater number of freeze–thaw cycles, for example, granite which is deteriorated after 136 freeze–thaw cycles (visible by variation of the apparent volume higher than 1%, after 256 cycles, the visual inspection indicates visible damages equal to 3 in the scale from 0 to 5, decrease on the dynamic modulus of elasticity close to 30%.

In the case of this paper, the freeze–thaw behavior was studied for the specimens with the maximum efficiency and for the simulated stone samples treated with 0.25 g/L solutions, respectively. In Table 4, the aspect before and after the freeze–thaw test and the gelivity coefficient could be visualized. For the untreated sample, the mechanical breaks can be observed, along with deep grooves, and shrinkage cracks due to the ice crystals formation. It has a higher porosity of the surface and a gelivity value $\mu_g$ of 9.56. The Mg-Sr-Zn-CHAp coating protects the stone specimens against the deterioration caused by consecutive freezing and thawing, and at the test end, no significant changes were observed. Although some pores appear on the surface, the gelivity coefficient has values smaller with 28% for the sprayed sample and 35% for the brushed sample, confirming a good consolidation of the treated stone specimens. A smaller gelivity coefficient highlights a smaller degradation rate and thus a higher protection capacity of Mg-Sr-Zn-CHAp applied on the stone specimens, as this paper proposed. The values can be also correlated with the higher consolidation percent resulting after the scotch tape test.

**Table 4.** The aspect and the gelivity coefficient after the freeze–thaw test of simulated stone samples treated with 0.25 g/L Mg-Sr-Zn-CHAp.

| Sample | Aspect before Test | Aspect after Test | Gelivity Coefficient, % |
|---|---|---|---|
| Untreated | | | 9.56 |
| Mg-Sr-Zn-CHAp Spraying (0.25g/L) | | | 6.89 |
| Mg-Sr-Zn-CHAp Brushing (0.25g/L) | | | 6.21 |

The compressive strength measured for the stone specimens subjected to 10 cycles of freeze–thaw tests was measured by the hammer impact energy test. As expected, the untreated specimen with visible degradation has a decreased compressive strength, the values of 3.86 MPa, being with 39.88% lower than the initial sample. For the specimens treated with 0.25 g/L of consolidant, as shown in Figure 6b, the superficial changes of treated samples lead to a decrease of 12.04% for spraying and of 9.81% for brushing, suggesting an adequate consolidation process.

The aesthetical changes produced by the consolidant application must be investigated in order to appreciate if the treatment is convincing enough. Thus, the chromatic parameters of the stone sample treated with Mg-Sr-Zn-CHAp in different concentrations were measured, as shown in Figure 7a,b.

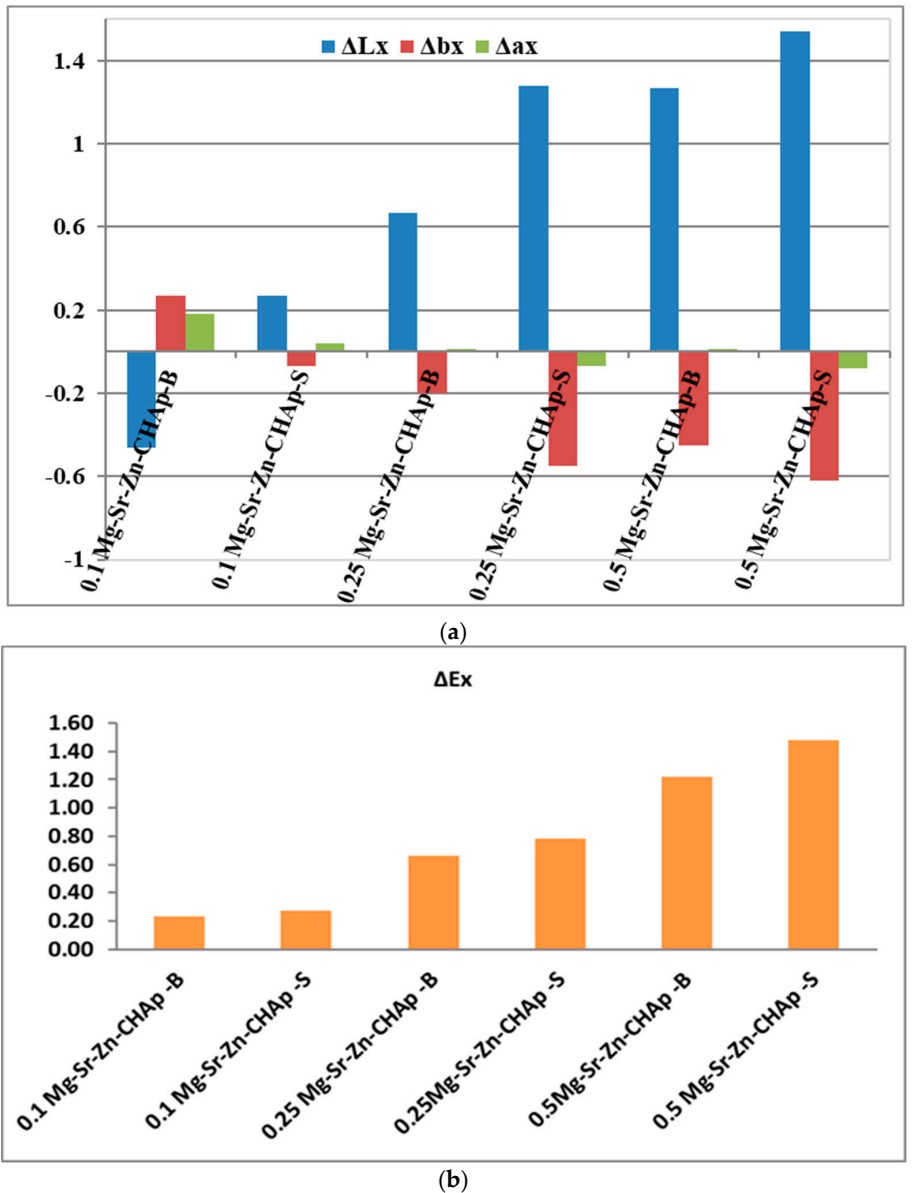

(a)

(b)

**Figure 7.** The variation of (**a**) chromatic parameters (ΔL*, Δa*, Δb*) and (**b**) the total color change (ΔE*) of the consolidated stone specimen samples.

Chromatic aberration is an important indicator of stone conservation. The coatings increase the grayscale, which is probably due to the white color of Mg-Sr-Zn-CHAp and does not have a great influence on the color.

According to other studies [9,70], the treatment with a consolidant can lead to a darkening of the sample, by reducing the luminosity, $L_x^*$, and yellowing when the $b_x^*$ parameter increases. The color of the untreated simulated stone sample falls within $\Delta L_x = 50.95$, $\Delta a_x = -2.79$ and $\Delta b_x = 5.74$. The values of ΔL* (as the difference between treated and untreated samples) show the change in lightness, positive values indicating that treated stone samples reflect the light more than the control, for all the analyzed samples. For brushed specimens, the change in lightness had values from 0.16 to 1.27, and for sprayed samples from 0.27 to 1.54, depending on the concentration. The yellowness decreases with increasing the consolidate concentration into the applied solution and slightly smaller values are observed for brushed samples. No significant changes in the red-green parameter are evidenced. Although chromatic parameters have changed values, these are all

between +1.48 and −0.62. In conclusion, the color changes are not significant and $\Delta E_x{}^*$ was calculated to emphasize this observation.

A decisive step in choosing a suitable consolidation treatment is the limit of the parameter $\Delta E_x^*$. Only solutions that impart an $\Delta E^* \leq 5$ are usually considered as acceptable [71,72], but other values could be taken into account, considering that the variations are observed by the naked eye when $\Delta E^* > 3$ [6]. Also, aesthetical compatibility of the treatments, according to the study by Delgado Rodrigues and co-workers [2], provided a useful classification system, in which they defined color variations on stone surfaces according to their risk of incompatibility, as follows: $\Delta E^* < 3$ risk of incompatibility low; $3 > \Delta E^* < 5$ medium incompatibility risk and $\Delta E^* > 5$ high incompatibility risk [73]. Considering the color changes of the treated specimens displayed in Figure 7, $\Delta E$ is lower than the threshold of the JND (Just Noticeable Difference) in the CIE Lab space ($\Delta E = 2.3$) [74]. The total color change increases with the consolidate's concentration and reaches 1.22 for brushed samples and 1.48 for sprayed simulated stone samples. $\Delta E_x$ values smaller than three show the compatibility of Mg-Sr-Zn-CHAp with the stone substrate, as the change is not detected by the human eye [33], sustaining the further use of the synthesized powder as consolidate for real stone samples.

By analyzing the results, it can be stated that triple-substituted carbonated hydroxyapatite applied especially by brushing on simulated stone samples ensures adequate cohesion with the substrate, improving the water absorption by blocking the surface pores of the stone and consolidating the samples as higher estimated compressive strength and improved resistance to freeze–thaw degradation result after treatment, without modifying the aesthetic aspect of the stones.

## 4. Conclusions

In this work, a new triple-substituted hydroxyapatite with three different metal ions, $Mg^{2+}$, $Sr^{2+}$ and $Zn^{2+}$, respectively were investigated and obtained in a single synthesis step, through the nanoemulsion technique, at room temperature. The resulting sample of Mg-Sr-Zn-CHAp was structurally, compositionally and morphologically characterized.

For the synthesized compound, the ratio of metal ions substituting calcium and $Ca^{2+}$ to phosphorus is 2.11, higher than for CHAp and pure hydroxyapatite.

The apatite structure was confirmed by XRD, which confirmed a B-form of apatite, and the component minerals have been identified by the Rietvelt refinement technique. This structure was also confirmed by FTIR. The incorporation of metal ions leads to small shifts of 1120–900 $cm^{-1}$ and absorption peaks of 500–600 $cm^{-1}$ (due to the metals incorporated into the CHAp structure).

Mg-Sr-Zn-CHAp has a granular structure, with spherical particle sizes between 150 and 250 nm, with a tendency to form agglomerates due to physical interactions between the particles.

The obtained powder was tested as a consolidant for stone specimens that were treated by both spraying and brushing with three solutions of different concentrations. The efficacy was demonstrated by improved water absorption, increased mechanical strength, higher percent consolidation and decreased freeze–thaw degradation rate. Also, no significant color changes were observed in the appearance of the treated samples.

Future research will evaluate the application of the Mg-Sr-Zn-CHAp treatment on real stones and the correlation of the results obtained for different stone substrates.

**Author Contributions:** Conceptualization, R.-M.I.; R.M.G. and L.I.; methodology, R.-M.I.; software, M.E.D.; validation, R.-M.I., L.I.; and R.M.G.; formal analysis, M.E.D.; investigation, L.I., R.M.G., R.-M.I., M.E.D., R.-M.I., A.I.G., E.A., L.P.; resources, R.-M.I.; data curation, L.I., R.M.G.; writing—original draft preparation, R.-M.I.; writing—review and editing, M.E.D.; L.I.; and R.-M.I.; visualization, R.-M.I.; supervision, R.-M.I.; project administration, R.-M.I.; funding acquisition, R.-M.I. All authors have read and agreed to the published version of the manuscript.

**Funding:** This research was funded by Ministry of Research, Innovation and Digitization, through the PN 23.06 Core Program—ChemNewDeal within the National Plan for Research, Development and Innovation 2022–2027, project No. PN 23.06.02.01 (InteGral) and project number PN-III-P2-2.1-PED-2021-3885 (687PED/2022) from UEFISCDI-MCID, within PNCDI III.

**Institutional Review Board Statement:** Not applicable.

**Informed Consent Statement:** Not applicable.

**Data Availability Statement:** Not applicable.

**Conflicts of Interest:** The authors declare that they have no known competing financial interest or personal relationships that could have appeared to influence the work reported in this paper.

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
