# Peer review of "New Triple Metallic Carbonated Hydroxyapatite for Stone Surface Preservation"

_coatings, doi:10.3390/coatings13081469_

Round 1

Reviewer 1 Report

Seem:

a) The work has scientific relevance and merit and is very interesting because it is research on the synthesis of carbonated hydroxyapatite being triple substituted by magnesium, strontium, and zinc (Mg-Sr-Zn-CHAp), and its structural, morphological, and compositional characteristics. For the work to be published, some modifications need to be made.

b) Summary meet the objectives of the work;

c) The introduction and review of the literature meet the objectives of the work;

d) Methodology is well written;

e) Tables 3 and 4'- They need to be better discussed with data from the literature of the last 05 years;

f) Figures 3, 5 and 8 - They need to be better discussed with data from the literature of the last 05 years;

g) Conclusion - Long conclusion and needs to be reduced to meet the objectives of the work;

h) This reviewer understands that the work should be published after modifications.

Author Response

Dear Reviewer,

Kind thanks for reviewing our paper in order to improve its quality.

Please find below our answers at your requests:

  1. The work has scientific relevance and merit and is very interesting because it is research on the synthesis of carbonated hydroxyapatite being triple substituted by magnesium, strontium, and zinc (Mg-Sr-Zn-CHAp), and its structural, morphological, and compositional characteristics. For the work to be published, some modifications need to be made.

Answer: Many thanks for your appreciations.

  1. Summary meet the objectives of the work;

Answer: We appreciate your opinion.

  1. The introduction and review of the literature meet the objectives of the work;

Answer: We appreciate your opinion.

  1. Methodology is well written;

Answer: We appreciate your opinion.

  1. Tables 3 and 4'- They need to be better discussed with data from the literature of the last 05 years;

Answer: Done. Supplementary paragraphs have been added in the text with corresponding references from the last 5 years.

  1. Figures 3, 5 and 8 - They need to be better discussed with data from the literature of the last 05 years;

Answer: Done. Supplementary paragraphs have been added in the text with corresponding references from the last 5 years.

  1. Conclusion - Long conclusion and needs to be reduced to meet the objectives of the work;

Answer: The Conclusions chapter has been shortened.

  1. This reviewer understands that the work should be published after modifications.

Answer: Thank you.

Prof.Rodica Mariana Ion

Author Response

Dear Reviewer,

Kind thanks for your efforts to review our paper and to improve its quality.

Please find below our answer at your evaluation form:

Summary

The manuscript submitted by Ramona Marina Grigorescu et. al. describes in detail the synthesis and its structural, morphological and composition of carbonated hydroxyapatite triple substituted with Mg, Sr and Zn. The paper fits very well the journal Coatings, although the literature is full of article concerning hydroxyapatite substitute. I appreciate the synthesis by nanoemulsion method, and all the experiments are presented very well. An accurate study is reported to justify the morphological and structural properties.  In my opinion, the manuscript draft is well presented and well organized, ready to be published.

Answer: We appreciate your kind evaluation and your answer.

Sincerely yours,

Prof.Rodica Mariana Ion

Reviewer 3 Report

I have made several comments in the attached manuscript and highlighted them with yellow color. Have a look!

Need to check the whole paper so that the long sentences could be made easily readable.

Author Response

Dear Reviewer,

Kind thanks for your efforts to review our paper and to improve its quality.

Please find below our answers at your requests:

Pg 1 line 3 Title need to be concise and attractive

Answer: We changed the title as follow:

New triple metallic carbonated hydroxyapatite for stone surface preservation

Pg 1 line 22 Long sentence with no clear message. Make it shorter and more meaningful.

Answer: Done

Pg 1 line 22 The whole abstract need to be re-written with some numerical findings and conclusions.

Answer: The abstract has been changed as it is presented below:

Abstract: This paper presents the synthesis of the triple substituted carbonated hydroxyapatite with magnesium, strontium and zinc (Mg-Sr-Zn-CHAp), as well as its structural, morphological and compositional characterization. The analytical techniques used (WDXRF, XRD and FTIR) highlighted, on the one hand, the B form for the apatite structure, as well as the presence of the three metal ions in the apatite structure, on the other hand (small shifts of 1120-900 cm-1 and 500-600 cm-1 absorption peaks (due to the metals incorporated into the CHAp structure)). The ratio between the metallic ions that substitute calcium and Ca2+, and phosphorus is increased, the value being 2.11 in comparison with CHAp and pure hydroxyapatite. Also, by using imaging techniques such as: optical microscopy and SEM, spherical nanometric particles (between 150 and 250 nm) with a large surface area and large pores (6 m2/ g surface area, pores with 6.903 nm diameters and 0.01035 cm3/g medium volume, determined by nitrogen adsorption/desorption analysis), and a pronounced tendency of agglomeration, were highlighted. Also, the triple substituted carbonated hydroxyapatite was tested as an inorganic consolidant, by using stone specimens prepared in the laboratory. The efficiency of Mg-Sr-Zn-CHAp in the consolidation processes was demonstrated by specific tests in the field: water absorption, exfoliation, freeze-thaw behavior, chromatic parameters as well as mechanical strength. All these tests presented conclusive values for the use of this consolidant in the consolidation procedures of stone surfaces (lower water absorption, increased mechanical strength, higher consolidation percent, decreased degradation rate by freeze-thaw, no significant color changes).

Pg 3 line 127 The whole experimental part needs to be re-written. I did not find any novelty in this part. Please make it concise and shorter. Also, show your contributions/novelty.

Answer: The equipment is not new. But they are mandatory in every paper in order to identify qualitatively and quantitively the prepared compounds. The Materials section has been shortened accordingly.

Pg 3 line 127 Make shorter sentences.

Answer: Done

Pg 6 line 219 Figure 2 needs more explanations. What is the significance of this figure? Please elaborate and show the scholarly findings of this figure.

Answer: We redraw the entire figure, and put into evidence the main aspects from this figure.

Pg 9 line 297 Title need to be more meaningful.

Answer: We reconsidered and corrected accordingly.

Pg 11 line 336 a, b, c, d are not clear, please mark them clearly.

Answer: Done

Pg 12 line 367 Loose sentence, rewrite please.

Answer: Rewritten.

Pg 16 line 491 Very very long sentence. Whole paper did not discuss any economic aspects. Then how you can call the finding is economic? Please either justify the or delete the economic term.

Answer: We deleted the economical term.

Sincerely yours

Prof.Rodica Mariana Ion

Round 2

Reviewer 3 Report

Looks ok to me!